# Synthesis of Flavone Derivatives via *N*-Amination and Evaluation of Their Anticancer Activities

**DOI:** 10.3390/molecules24152723

**Published:** 2019-07-26

**Authors:** Ni Zhang, Jin Yang, Ke Li, Jun Luo, Su Yang, Jun-Rong Song, Chao Chen, Wei-Dong Pan

**Affiliations:** 1State Key Laboratory of Functions and Applications of Medicinal Plants, Guizhou Medical University, Guiyang 550014, China; 2The Key Laboratory of Chemistry for Natural Products of Guizhou Province and Chinese Academy of Sciences/Guizhou Provincal Engineering Research Center for Natural Drugs, Guiyang 550014, China; 3College of Pharmacy, Zunyi Medical University, Zunyi 563099, China

**Keywords:** flavone, amino acid, anticancer activity, antiproliferation

## Abstract

Seventeen new flavone derivatives substituted at the 4′-OH position were designed, synthesized and evaluated for their anticancer and antibacterial activities. Among them, compounds **3**, **4**, **6f**, **6e**, **6b**, **6c** and **6k** demonstrated the most potent antiproliferative activities against a human erythroleukemia cell line (HEL) and a prostate cancer cell line (PC3). The results also showed that the IC_50_ value of compounds **3**, **4**, **6f**, **6e**, **6b**, **6c** and **6k** were close to that of the anticancer drug cisplatin (DDP) and lower than that of apigenin. All of the derivatives did not present antibacterial activities. The structure–activity relationships evaluation showed that the configuration of methyl amino acid might affect their biological activities.

## 1. Introduction

Cancer is ranked as the main cause of human death and the most important barrier to increasing life expectancy around the world. There were 18.1 million new cases and 9.6 million cancer deaths worldwide in 2018 [1]. Current cancer treatments mainly rely on surgery, chemotherapy, and radiotherapy, which cannot reduce recurrence and metastasis, as well as other limitations and drawbacks, such as severe side-effects, intolerance and increasing resistance. Hence, the need for developing new anticancer agents is growing [2].

Nowadays, phytochemicals have become an important part of anticancer drugs. Actually, over 75% of nonbiological anticancer drugs approved are either natural products or developed based on them [3]. Flavonoids, which are polyphenolic secondary metabolites mainly from plants and fungi, have diverse biological activities—especially anticancer activity through the regulation of different targets. For instance, some research results have shown flavonoids targeting protein kinase (PKC) [4], tankyrase (TNKS) [5], tyrosine kinases [6] and cyclin dependent kinases [7]. Therefore, flavone was chosen as the lead skeleton for further structural modification for discovering new anticancer drug candidates. 

As far back as 1986, Zhao et al. [8] studied the structure–activity relationships of flavone and found that 4′-OH was important for inhibitory activity. Golub et al. [9] reported the design, synthesis and characterization of novel flavone 4′-OH derivatives, which showed better anticancer activities than the natural lead compound. Meanwhile, the 4′-OH could form a hydrogen bond with Asp175 and stacking interaction with Phe113, which further enhances their activities [9]. Additionally, Zhou et al. [10] designed a series of flavone derivatives with alkanes substituted in 4′-OH to generate ether. 

The lipophilicity and hydrophilicity of an amino acid may have a large influence on the ester–water distribution coefficient of the target molecule, thereby retaining the possibility of a hydrogen bond interaction. Biological activity results showed that flavone binds the methyl amino acid at the 4′-position, which can increase its anticancer activities. Meanwhile, two alkyl amines (propylamine and n-butylamine) and 2-fluoroaniline were chosen to bond to flavone for comparison with methyl amino acid. Based on this analysis, a series of flavone derivatives were designed and synthesized by replacing the 4′-OH with *N*-amino substituents.

## 2. Results and Discussion

### 2.1. Chemistry

Seventeen new flavone derivatives (**3**, **4**, **6**, **6a**–**6n**) were synthesized (Scheme 1). To begin with, commercially available 2,4,6-trihydroxyacetophenone was methylated with MeI in dried DMF (Scheme 2), resulting in the desired product in 45% yield. Then, the synthetic method was changed by using Me_2_SO_4_ as the methylation reagent instead of MeI, and only the target product was obtained in 98% yield [11]. The mixture was subsequently filtered and the pure product was obtained after removing the solvent under vacuum distillation (without further purification). By following a method analogous to that described in Reference [12,13], **2b** was easily obtained. The intermediate chalcone **3** was achieved via aldol condensation between **1a** and **2b** at 40 °C with NaOH (aq 40%) as the base [14,15]. The temperature and NaOH concentration were the key issues to this transformation. Chalcone **3** was cyclized with I_2_ as a catalyst at 160 °C in DMSO to form compound **4** [15]. Compound **5** was synthesized by a known method with Pd/C (10%) as catalyst in MeOH [16]. With excess K_2_CO_3_ as a base, compound **6** could be prepared from **5** [17,18]. Compounds **6a**–**6n** were synthesized from various amines and compound **6** by palladium catalyzed *N*-arylation reactions [17]. In summary, 17 new flavone derivatives (**3**, **4**, **6**, **6a**–**6n**) were synthesized via several steps (Scheme 1).

### 2.2. Anticancer Bioactivity 

The 17 flavone derivatives were tested against HEL and PC3 cell lines. The IC_50_ values of the flavone derivatives against the growth of HEL and PC3 cell lines are shown in Table 1 and Figure 1.

From the MTT assay after 48 h of treatment; the values are averaged from at least three independent experiments; variation ±10%.

Both apigenin and the anticancer drug cisplatin (DDP) were chosen as positive controls. As depicted in Table 1, compounds **3**, **4**, **6f**, **6e**, **6b**, **6c**, and **6k** exhibited more potent effects against the growth of HEL and PC3 cell lines compared to that of apigenin. The results showed that **6e**, **6b** and **6k** had better anticancer activities than the positive control DDP (Figure 1). Interestingly, the anticancer activities of compounds **3** and **4**, which were protected with benzyloxy at 4′-OH, were greater than that of **5**. This means it makes sense to modify the structure of flavone at 4′-OH. Four methyl amino acids (methyl l-leucinate, methyl d-leucinate, l-valine methyl ester, and d-valine methyl ester) were chosen to bond to flavone by comparing their activities against the growth of cancer cells. We found the activity of **6f** was better than that of **6g**, while **6b** and **6c** (D and L configuration, respectively) had similar IC_50_ values. These results indicate that the configuration of the methyl amino acid may have an effect on the biological activities.

The cell growth curves following treatment by compound **6k** are presented in Figure 2. Cell morphology was significantly changed after drug treatment; an increase in concentration and duration of **6k** treatment led to an increase in the inhibition rate of cell growth for HEL and PC3 cells. When the time of drug treatment reached 36 h, the OD (absorbance values) value decreased slowly, meaning that the inhibition rate of cell growth increased gradually. This indicated that the biological activities of the flavone derivatives were dependent on time and concentration.

Meanwhile, the 17 flavone derivatives were also tested against numerous bacteria species (*Bacillus subtilis*, *Pseudomonas aeruginosa*, *anthraci*, *Staphylococcus aureus* 6538, *Staphylococcus aureus* 43300, *Staphylococcus aureus* 25923, and *Escherichia coli*). The results showed that the derivatives did not have antibacterial activities.

## 3. Materials and Methods

### 3.1. Instruments and Materials

Reagents and solvents were purchased from commercial sources. Solvents were purified according to the guidelines in Purification of Laboratory Chemicals. Column chromatography was performed on silica gel (Huang Hai, 200–300 mesh) using the indicated eluents. A stock solution of **6k** (20 µM) was prepared in DMSO (dimethyl sulfoxide) and stored in the refrigerator at −20 °C. MTT (3-(4,5-dimethyl-2-thiazolyl)-2,5-diphenyl-2-*H*-tetrazolium bromide or thiazolyl blue tetrazolium bromide) was purchased from Solarbio (Shang Hai, China). Melting points were measured on SGW X-4 apparatus and were not corrected. ^1^H-NMR and ^13^C-NMR spectra were recorded on 400 MHz (Varian, Palo Alto, CA, USA) or 600 MHz (Bruker, Karlsruhe, Germany) spectrometers in appropriate solvents using TMS (tetramethylsilane) as the internal standard or the solvent signals as secondary standards. Multiplicities of the NMR signals were designated as s (singlet), d (doublet), t (triplet), q (quartet), br (broad), and m (multiplet). High-resolution mass spectra(HRMS) were obtained using Thermo Fisher QE Focus apparatus(USA). 

### 3.2. Methods of Synthesis

#### 3.2.1. Synthesis of 2,4-Dimethoxy-3-hydroxyacetophenone **1a**

After K_2_CO_3_ (10.0 mmol, 2.0 eq) was added to a solution of 2,4,6-trihydroxyacetophenone **1** (5.0 mmol, 1.0 eq), the resulting solution was stirred in acetone at room temperature under argon. Then, Me_2_SO_4_ (10.0 mmol, 2.0 eq) was added dropwise and the temperature was slowly increased to 40 °C. The reaction was monitored by thin layer chromatography (TLC) after about 4 h. The mixture was then filtered and washed with acetone three times. The organic phase was evaporated to provide **1a** (white powder; yield 98%). Other data can be found in Reference [11].

#### 3.2.2. Synthesis of 3,5-Dimethyl-4-benzyloxybenzaldehyde **2b**

K_2_CO_3_ (12.0 mmol, 2.0 eq) was added to a solution of 3,5-dimethyl-4-hydroxybenzaldehyde **2** (6.0 mmol, 1.0 eq) in DMF (dimethyl formamide) (40 mL), which was stirred at 0 °C. Then, BrBn (6.6 mmol, 1.1 eq) was added dropwise and the mixture was allowed to increase in temperature to room temperature. The end of reaction was monitored by TLC about 4.5 h, then water was added (50 mL). The solution was then extracted with EA (ethyl acetate) (20 mL × 3).The combined organic phase was washed with water (20 mL × 3) and brine (20 mL × 3), and dried over anhydrous Na_2_SO_4_ after filtration. The solvent was removed under reduced pressure to get the crude product, which was purified through flash column chromatography to afford compound **2b** (white solid; yield 95%). Other data can be found in Reference [13].

#### 3.2.3. Synthesis of 2′,4′-Dimethoxyl-5′-hydroxy-3,5-dimethyl-4-benzyloxychalcone **3**

An aqueous solution of 40% NaOH (50 mmol, 10.0 eq) was added to a solution of **1a** (5.0 mmol, 1.0 eq) and **2b** (5.0 mmol, 1.0 eq) in EtOH (40 mL) at room temperature. After stirring for 4 h at 40 °C, the mixture was adjusted with 1 N HCl to pH 5 and filtered. The precipitate was recrystallized with EtOH to afford **3** (yellow solid; yield 70%). Melting point (mp): 120.6–121.4 °C; Log *P*: 5.52; ^1^H-NMR (400 MHz, CDCl_3_) *δ*(ppm): 14.40 (s, 1H, OH), 7.82 (d, *J* = 15.6 Hz, 1H, C=CH), 7.73 (d, *J* = 15.6 Hz, 1H, C=C–H), 7.50 (d, *J* = 6.8 Hz, 2H, Ar–H), 7.37–7.45 (m, 3H, Ar–H), 7.30 (s, 2H, Ar–H), 6.12 (d, *J* = 2.4 Hz, 1H, Ar–H), 5.98 (d, *J* = 2.4 Hz, 1H, Ar–H), 4.85 (s, 2H, ArCH_2_), 3.93 (s, 3H, OCH_3_), 3.84 (s, 3H, OCH_3_), 2.34 (s, 6H, ArCH_3_); ^13^C-NMR (100 MHz, CDCl_3_) *δ*(ppm): 192.6, 168.3, 166.0, 162.4, 157.6, 142.4, 137.2, 131.7, 131.2, 129.2, 128.5, 128.1, 127.8, 126.3, 106.3, 93.7, 91.2, 74.1, 55.9, 55.6, 16.6; HRMS (ESI) calcd. C_26_H_27_O_5_: *m*/*z* 419.1853 [M + H]^+^, found: 419.1850.

#### 3.2.4. Synthesis of 5,7-Dimethoxy- 3′,5′-dimethyl-4′-benzyloxyflavone **4**

A mixture of **3** (2.4 mmol, 1.0 eq) and iodine (0.024 mmol, 0.01 eq) in DMSO (30 mL) was refluxed for 4 h, then cooled down to room temperature and poured into water. The crude product was obtained after filtration and recrystallized in EA to get product **4** (white solid; yield 72%). Mp: 208.1–209.2 °C; Log *P*: 5.4; ^1^H-NMR (400 MHz, CDCl_3_) *δ*(ppm): 7.56 (s, 2H, Ar–H), 7.50 (d, *J* = 4.0 Hz, 2H, Ar–H), 7.43 (t, *J* = 4.0 Hz, 2H, Ar–H), 7.39 (t, *J* = 4.0 Hz, 1H, Ar–H), 6.62 (s, 1H, C=CH), 6.58 (d, *J* = 1.2 Hz, 1H, Ar–H), 6.37 (d, *J* = 1.2 Hz, 1H, Ar–H), 4.87 (s, 2H, ArCH_2_), 3.96 (s, 3H, OCH_3_), 3.93 (s, 3H, OCH_3_), 2.37 (s, 6H, ArCH_3_); ^13^C-NMR (151 MHz, CDCl_3_) *δ*(ppm): 177.7, 164.0, 160.9, 160.7, 159.9, 158.5, 137.1, 132.0, 128.6, 128.2, 127.9, 127.0, 126.7, 109.2, 108.5, 96.1, 92.8, 74.2, 56.4, 55.8, 16.7; HRMS (ESI) calcd. C_26_H_25_O_5_: *m*/*z* 417.1697 [M + H]^+^, found: 417.1695. 

#### 3.2.5. Synthesis of 5,7-Dimethoxy- 3′,5′-dimethyl-4′-hydroxyflavone **5**

Compound **4** (2.4 mmol, 1.0 eq) and Pd/C (10%, 0.24 mmol, 0.1 eq) were added to a two-neck flask. The air in the flask was then exchanged by hydrogen after MeOH (50 mL) was added. The mixture was stirred at room temperature for 4 h before it was filtered and washed with heated MeOH (20 mL × 3). The MeOH was removed under reduced pressure to get product **5** (white solid, yield 95%). Mp: 206.3–207.1 °C; Log *P*: 3.4; ^1^H-NMR (600 MHz, DMSO-*d_6_*) *δ*(ppm): 9.03 (s, 1H, OH), 7.64 (s, 2H, Ar–H), 6.84 (d, *J* = 0.4 Hz, 1H, Ar–H), 6.57 (s, 1H, C=CH), 6.48 (d, *J* = 1.2 Hz, 1H, Ar–H), 3.9 (s, 3H, OCH_3_), 3.82 (s, 3H, OCH_3_), 2.24 (s, 6H, ArCH_3_); ^13^C-NMR (151 MHz, DMSO-*d_6_*) *δ*(ppm): 176.1, 164.0, 160.8, 160.6, 159.6, 157.0, 126.7, 125.2, 121.7, 108.7, 106.6, 96.7, 93.7, 56.5, 56.4, 17.1; HRMS (ESI) calcd. C_19_H_19_O_5_N: *m*/*z* 327.1227 [M + H]^+^, found: 327.1226.

#### 3.2.6. Synthesis of 5,7-Dimethoxy-3′,5′-dimethyl-4′-trifluoromethanesulfonyloxyflavone **6**

Compound **5** (3.0 mmol, 1.0 eq), K_2_CO_3_ (30.0 mmol, 10.0 eq) and anhydrous DMF (40 mL) were added into a two-neck flask under argon. After the mixture was cooled to 0 °C in an ice bath, trifluoromathanesulfonyl chloride was added dropwise. The reaction mixture was slowly warmed to room temperature and stirring was continued for 6 h before it was filtered and then poured into water (50 mL) and extracted with CH_2_Cl_2_ (20 mL × 3). The combined organic layers were washed with brine (20 mL), dried over anhydrous Na_2_SO_4_ and concentrated in a vacuum to get a crude product. The crude product was chromatographed on silica gel (EA/PE = 2/1) to get **6** (white solid; yield 60%). Mp: 265.5–266.2 °C; Log *P*: 4.87; ^1^H-NMR (600 MHz, CDCl_3_) *δ*(ppm): 7.60 (s, 2H, Ar–H), 6.62 (s, 1H, C=CH), 6.57 (d, *J* = 2.4 Hz, 1H, Ar–H), 6.37 (d, *J* = 1.8 Hz, 1H, Ar–H), 3.95 (s, 3H, OCH_3_), 3.92 (s, 3H, OCH_3_), 2.47 (s, 6H, CH_3_); ^13^C-NMR (151 MHz, CDCl_3_) *δ*(ppm): 177.2, 164.2, 160.9, 159.8, 159.0, 148.4, 132.5, 131.2, 127.4, 118.6 (q, *J* = 320.1 Hz, CF_3_) 109.7, 109.2, 96.3, 92.8, 56.4, 55.8, 17.4; HRMS (ESI) calcd. C_20_H_18_O_7_F_3_S: *m*/*z* 459.0720 [M + H]^+^, found: 459.0721.

#### 3.2.7. Synthesis of **6a**–**6n**

Toluene (10 mL) was added to a mixture of palladium acetate (0.02 mmol, 0.1 eq), BINAP((±)-2,2′-Bis(diphenylphosphino)-1,1′-binaphthalene) (0.04 mmol, 0.2 eq), cesium carbonate (0.4 mmol, 2.0 eq), **6** (0.22 mmol, 2.2 eq) and amine (0.2 mmol, 1.0 eq) under argon. The mixture was stirred at reflux for 12 h. After cooling to room temperature and removing solvents, the mixture was extracted with DCM (dichloromethane) and washed with brine. The organic layer was dried over Na_2_SO_4_, followed by purification of the crude product by column chromatography on silica gel (DCM/MeOH = 100/1). This afforded compounds **6a**–**6n**. 

**6a:***Methyl (4-(5,7-dimethoxy-4-oxo-4H-chromen-2-yl)-2,6-dimethylphenyl)glycinate*: Yellow powder; yield 60%; mp: 134.3–135.7 °C; Log *P*: 2.82; ^1^H-NMR (400 MHz, CDCl_3_) *δ*(ppm): 7.49 (s, 2H, Ar–H), 6.56 (s, 2H, Ar–H and C=CH), 6.35 (s, 1H, Ar–H), 4.26 (s, 1H, NH), 3.94 (s, 5H, OCH_3_ and COCH_2_), 3.91 (s, 3H, OCH_3_), 3.77 (s, 3H, OCH_3_), 2.38 (s, 6H, ArCH_3_); ^13^C-NMR (151 MHz, CDCl_3_) *δ*(ppm): 177.7, 172.4, 163.8, 161.1, 160.8, 159.9, 148.9, 127.6, 126.8, 123.7, 109.2, 107.4, 96.0, 92.8, 56.4, 55.7, 52.4, 49.3, 19.1; HRMS (ESI) calcd. C_22_H_24_O_6_N: *m*/*z* 398.1598 [M + H]^+^, found: 398.1596.

**6b:***Methyl (4-(5,7-dimethoxy-4-oxo-4H-chromen-2-yl)-2,6-dimethylphenyl)-l-leucinate*: Yellow powder; yield 55%; mp: 192.7–193.3 °C; Log *P*: 4.55; ^1^H-NMR (400 MHz, CDCl_3_) *δ*(ppm): 7.47 (s, 2H, Ar–H), 6.55–6.56 (m, 1H, Ar–H and C=CH), 6.34 (d, *J* = 2.0 Hz, 1H, Ar–H), 6.36 (d, *J* = 2.0 Hz, 1H, Ar–H), 4.10–4.15 (m, 1H, NHC*H*), 3.93 (s, 3H, OCH_3_), 3.90 (s, 3H, OCH_3_), 3.61 (s, 3H, OCH_3_), 2.36 (s, 6H, ArCH_3_), 1.61–1.80 (m, 3H, CH_2_CH), 0.97 (t, *J* = 7.2 Hz, 6H, CH_3_); ^13^C-NMR (100 MHz, CDCl_3_) *δ*(ppm): 177.7, 175.1, 163.7, 161.0, 160.8, 159.8, 147.4, 128.1, 126.8, 123.8, 109.2, 107.4, 95.9, 92.7, 57.7, 56.3, 55.7, 51.9, 43.6, 24.9, 22.8, 22.5, 19.2; HRMS (ESI) calcd. C_26_H_32_O_6_N: *m*/*z* 454.2224 [M + H]^+^, found: 454.2223.

**6c:***Methyl (4-(5,7-dimethoxy-4-oxo-4H-chromen-2-yl)-2,6-dimethylphenyl)-d-leucinate*: Yellow powder; yield 59%; mp: 166.1–167.4 °C; Log *P*: 4.55; ^1^H-NMR (400 MHz, CDCl_3_) *δ*(ppm): 7.47 (s, 2H, Ar–H), 6.57 (s, 1H, C=CH), 6.56 (d, *J* = 2.0 Hz, 1H, Ar–H), 6.35 (d, *J* = 2.0 Hz, 1H, Ar–H), 4.13 (t, *J* = 6.8 Hz, 1H, NHC*H*), 3.93 (s, 3H, OCH_3_), 3.90 (s, 3H, OCH_3_), 3.61 (s, 3H, OCH_3_), 2.37 (s, 6H, ArCH_3_), 1.60–1.77 (m, 3H, C*H*_2_C*H*), 0.98 (t, *J* = 7.2 Hz, 6H, CH_3_); ^13^C-NMR (100 MHz, CDCl_3_) *δ*(ppm): 177.7, 175.1, 163.8, 161.1, 160.7, 159.8, 147.4, 128.1, 126.9, 123.8, 109.1, 107.3, 95.9, 92.7, 57.7, 56.3, 55.7, 51.9, 43.6, 24.9, 22.8, 22.5, 19.2; HRMS (ESI) calcd. C_26_H_32_O_6_N: *m*/*z* 454.2224 [M + H]^+^, found: 454.2223.

**6d:***Methyl (4-(5,7-dimethoxy-4-oxo-4H-chromen-2-yl)-2,6-dimethylphenyl)-l-methioninate*: Yellow powder; yield 40%; mp: 142.0–143.1 °C; Log *P*: 4.87; ^1^H-NMR (400 MHz, CDCl_3_) *δ*(ppm): 7.49 (s, 2H, Ar–H), 6.56–6.57 (m, 2H, Ar–H and C=CH), 6.36 (d, *J* = 4.0 Hz, 1H, Ar–H), 4.26 (s, 1H, NH), 3.96–4.09 (m, 1H, NHC*H*), 3.94 (s, 3H, OCH_3_), 3.91 (s, 3H, OCH_3_), 3.66 (s, 3H, OCH_3_), 2.64 (t, *J* = 8.0 Hz, 2H, SCH_2_), 2.39 (s, 6H, ArCH_3_), 2.09 (s, 3H, SCH_3_), 2.00–2.12 (m, 2H, CHC*H*_2_); ^13^C-NMR (100 MHz, CDCl_3_) *δ*(ppm):177.7, 174.4, 163.8, 161.0, 160.7, 159.8, 147.1, 128.2, 126.8, 123.9, 109.1, 107.4, 95.9, 92.7, 57.8, 56.3, 55.7, 52.1, 30.1, 29.6, 19.2, 15.4; HRMS (ESI) calcd. C_25_H_30_O_6_NS: *m*/*z* 472.1788 [M + H]^+^, found: 472.1791.

**6e:***Methyl (2S)-2-((4-(5,7-dimethoxy-4-oxo-4H-chromen-2-yl)-2,6-dimethylphenyl)amino)-3-methylpentanoate*: Yellow powder; yield 45%; mp: 151.3–153.4 °C; Log *P*: 5.84; ^1^H-NMR (600 MHz, CDCl_3_) *δ*(ppm): 7.43 (s, 2H, Ar–H), 6.53 (s, 1H, C=C–H), 6.52 (d, *J* = 1.8 Hz, 1H, Ar–H), 6.30 (d, *J* = 2.4 Hz, Ar–H), 4.01–4.05 (m, 2H, N*H*C*H*), 3.90 (s, 3H, OCH_3_), 3.86 (s, 3H, OCH_3_), 3.60 (s, 3H, OCH_3_), 2.34 (s, 6H, ArCH_3_), 1.78–1.85 (m, 1H, CH_3_C*H*), 1.68–1.72 (m, 1H, CH_3_C*H*_2_), 1.51-1.59 (m, 1H, CH_3_C*H*_2_), 0.90–0.97 (m, 6H, CHC*H*_3_ and CH_2_C*H*_3_); ^13^C-NMR (151 MHz, CDCl_3_) *δ*(ppm): 177.8, 174.2, 163.8, 161.2, 160.8, 159.8, 147.5, 127.8, 127.0, 123.5, 109.2, 107.2,, 96.0, 92.8, 63.2, 56.3, 55.7, 51.7, 39.0, 25.9, 19.3, 15.2, 11.7; HRMS (ESI) calcd. C_26_H_32_O_6_N: *m*/*z* 454.2224 [M + H]^+^, found: 454.2225.

**6f:***Methyl (4-(5,7-dimethoxy-4-oxo-4H-chromen-2-yl)-2,6-dimethylphenyl)-l-valinate*: Yellow powder; yield 55%; mp: 172.6–173.3 °C; Log *P*: 5.42; ^1^H-NMR (400 MHz, CDCl_3_) *δ*(ppm): 7.47 (s, 2H, Ar–H), 6.55-6.56 (m, 2H, Ar–H and C=CH), 6.34 (d, *J* = 2.0 Hz, 1H, Ar–H), 4.03–4.06 (m, 1H, COCH), 3.94 (s, 3H, OCH_3_), 3.90 (s, 3H, OCH_3_), 3.62 (s, 3H, OCH_3_), 2.37 (s, 6H, ArCH_3_), 2.04–2.12 (m, 1H, (CH_3_)_2_C*H*), 1.13 (d, *J* = 7.2 Hz, 3H, CH_3_), 1.01 (d, *J* = 7.2 Hz, 3H, CH_3_); ^13^C-NMR (100 MHz, CDCl_3_) *δ*(ppm): 177.7, 174.3, 163.7, 161.1, 160.7, 159.8, 147.4, 127.8, 126.9, 123.5, 109.2, 107.3, 95.9, 92.7, 64.5, 56.3, 55.7, 51.7, 32.2, 19.3, 18.9, 18.7; HRMS (ESI) calcd. C_25_H_30_O_6_N: *m*/*z* 440.2068 [M + H]^+^, found: 440.2065.

**6g:***Methyl (4-(5,7-dimethoxy-4-oxo-4H-chromen-2-yl)-2,6-dimethylphenyl)-d-valinate*: Yellow powder; yield 53%; mp: 166.8–168.4 °C; Log *P*: 4.2; ^1^H-NMR (400 MHz, CDCl_3_) *δ*(ppm): 7.47 (s, 2H, Ar–H), 6.55–6.56 (m, 2H, Ar–H and C=CH), 6.35 (d, *J* = 2.4 Hz, 1H, Ar–H), 4.03–4.06 (m, 1H, NHC*H*), 3.94 (s, 3H, OCH_3_), 3.92 (s, 3H, OCH_3_), 3.62 (s, 3H, OCH_3_), 2.37 (s, 6H, ArCH_3_), 2.04–2.12 (m, 1H, (CH_3_)_2_C*H*), 1.13 (d, *J* = 7.2 Hz, CH_3_), 1.01 (d, *J* = 7.2 Hz, CH_3_); ^13^C-NMR (100 MHz, CDCl_3_) *δ*(ppm): 177.7, 174.3, 163.7, 161.1, 160.7, 159.8, 147.4, 127.8, 126.9, 123.5, 109.0, 107.3, 95.9, 92.7, 64.5, 56.3, 55.7, 51.7, 32.2, 19.2, 18.9, 18.7; HRMS (ESI) calcd. C_25_H_30_O_6_N: *m*/*z* 440.2068 [M + H]^+^, found: 440.2066.

**6h:***Methyl (4-(5,7-dimethoxy-4-oxo-4H-chromen-2-yl)-2,6-dimethylphenyl)-l-alaninate*: Yellow powder; yield 55%; mp: 183.2–184.0 °C; Log *P*: 3.31; ^1^H-NMR (600 MHz, CDCl_3_) *δ*(ppm): 7.49 (s, 2H, Ar–H), 6.58 (s, 1H, C=CH), 6.56 (d, *J* = 1.8 Hz, 1H, Ar–H), 6.35 (d, *J* = 2.4 Hz, 1H, Ar–H), 4.15 (dd, *J* = 6.6, 13.8 Hz, 1H, COCH), 3.94 (s, 3H, OCH_3_), 3.90 (s, 3H, OCH_3_), 3.69 (s, 3H, OCH_3_), 2.37 (s, 6H, ArCH_3_), 1.42 (d, *J* = 7.2 Hz, 3H, CH_3_); ^13^C-NMR (151 MHz, CDCl_3_) *δ*(ppm): 177.8, 175.5, 163.9, 161.2, 160.9, 159.9, 147.4, 128.5, 126.9, 124.0, 109.2, 107.5, 96.0, 92.8, 56.4, 55.8, 54.7, 52.2, 19.9, 19.2; HRMS (ESI) calcd. C_23_H_26_O_6_N: *m*/*z* 412.1755 [M + H]^+^, found: 412.1753.

**6i:***Dimethyl (4-(5,7-dimethoxy-4-oxo-4H-chromen-2-yl)-2,6-dimethylphenyl)-l-glutamate*: Yellow powder; yield 10%; mp: 123.3–124.0 °C; Log *P*: 3.08; ^1^H-NMR (600 MHz, CDCl_3_) *δ*(ppm): 7.47 (s, 2H, Ar–H), 6.56 (s, 1H, C=CH), 6.55 (d, *J* = 2.4 Hz, 1H, Ar–H), 6.35 (d, *J* = 2.4 Hz, 1H, Ar–H), 4.14 (s, 1H), 3.94 (s, 3H, OCH_3_), 3.90 (s, 3H, OCH_3_), 3.69 (s, 3H, OCH_3_), 3.65 (s, 3H, OCH_3_), 2.49–2.55 (m, 2H, CHC*H*_2_), 2.36 (s, 6H, ArCH_3_), 2.10–2.14 (m, 2H, COC*H*_2_); ^13^C-NMR (151 MHz, CDCl_3_) *δ*(ppm): 177.8, 174.4, 173.2, 163.9, 161.1, 160.9, 159.9, 147.1, 128.4, 127.0, 124.1, 109.3, 107.6, 96.0, 92.8, 58.3, 56.4, 55.8, 52.2, 51.9, 30.1, 28.8, 19.2; HRMS (ESI) calcd. C_26_H_30_O_8_N: *m*/*z* 484.1966 [M + H]^+^, found: 484.1959.

**6j:***Methyl (4-(5,7-dimethoxy-4-oxo-4H-chromen-2-yl)-2,6-dimethylphenyl)-l-phenylalaninate*: Yellow powder; yield 60%; mp: 174.1–175.9 °C; Log *P*: 4.99; ^1^H-NMR (400 MHz, CDCl_3_) *δ*(ppm): 7.46 (s, 2H, Ar–H), 7.25–7.32 (m, 3H, Ar–H), 7.14 (d, *J* = 7.2 Hz, Ar–H), 6.57 (s, 1H, C=CH), 6.56 (d, *J* = 2.4 Hz, 1H, Ar–H), 6.35 (d, *J* = 2.0 Hz, 1H, Ar–H), 4.34 (t, *J* = 6.4 Hz, 1H, COCH), 3.93 (s, 3H, OCH_3_), 3.90 (s, 3H, OCH_3_), 3.60 (s, 3H, COOCH_3_), 3.10 (m, 2H, ArCH_2_), 2.23 (s, 6H, ArCH_3_); ^13^C-NMR (100 MHz, CDCl_3_) *δ*(ppm): 177.7, 174.1, 163.8, 161.0, 160.7, 159.8, 147.1, 136.1, 129.3, 128.4, 128.1, 127.0, 126.8, 123.7, 109.1, 107.3, 96.0, 92.7, 60.2, 56.3, 55.7, 51.9, 40.0, 19.1; HRMS (ESI) calcd. C_29_H_30_O_6_N: *m*/*z* 488.2068 [M+H]^+^, found: 488.2068.

**6k:***Methyl N^6^-(tert-butoxycarbonyl)-N^2^-(4-(5,7-dimethoxy-4-oxo-4H-chromen-2-yl)-2,6-dimethylphenyl)-l-lysinate*: Yellow powder; yield 60%; mp: 92.4–94.4 °C; Log *P*: 4.45, ^1^H-NMR (600 MHz, CDCl_3_) *δ*(ppm): 7.44 (s, 2H, Ar–H), 6.52–6.53 (m, 2H, Ar–H and C=CH), 6.31 (d, *J* = 1.8 Hz, 1H, Ar–H), 4.61 (s, 1H, NH), 4.05–4.07 (m, 1H, NHC*H*), 4.0 (br, 1H, CONH), 3.90 (s, 3H, OCH_3_), 3.87 (s, 3H, OCH_3_), 3.62 (s, 3H, OCH_3_), 3.08–3.10 (m, 2H, CONH*CH*_2_), 2.33 (s, 6H, CH_3_), 1.76–1.79 (m, 2H, COCH*CH*_2_), 1.36–1.51 (m, 4H, C*H*_2_C*H*_2_), 1.40 (s, 9H, C(CH_3_)_3_); ^13^C-NMR (151 MHz, CDCl_3_) *δ*(ppm): 177.7, 174.8, 163.8, 161.1, 160.8, 159.8, 156.0, 147.4, 128.2, 126.9, 123.8, 109.2, 107.4, 96.0, 92.8, 79.1, 59.0, 56.4, 55.7, 52.0, 40.2, 33.7, 29.9, 28.4, 22.8, 19.2; HRMS (ESI) calcd. C_31_H_41_O_8_N_2_: *m*/*z* 569.2857 [M + H]^+^, found: 569.2856.

**6l:***2-(3,5-dimethyl-4-(propylamino)phenyl)-5,7-dimethoxy-4H-chromen-4-one*: Yellow powder; yield 62%; mp: 151.5–152.6 °C; Log *P*: 4.11; ^1^H-NMR (600 MHz, CDCl_3_) *δ*(ppm): 7.49 (s, 2H, Ar–H), 6.54 (s, 2H, Ar–H and C=CH), 6.32 (d, *J* = 1.6 Hz, 1H, Ar–H), 3.91 (s, 3H, OCH_3_), 3.88 (s, 3H, OCH_3_), 3.09 (t, *J* = 7.2 Hz, 2H, NHC*H*_2_), 2.31 (s, 6H, ArCH_3_), 1.54–1.60 (m, 2H, CH_3_C*H*_2_), 0.96 (t, *J* = 7.2 Hz, 3H, CH_3_); ^13^C-NMR (151 MHz, CDCl_3_) *δ*(ppm): 177.8, 163.8, 161.3, 160.8, 159.9, 149.7, 127.5, 126.8, 122.8, 109.2, 107.1, 96.0, 92.8, 56.4, 55.7, 49.9, 24.4, 19.2, 11.5; HRMS (ESI) calcd. C_22_H_26_O_4_N: *m*/*z* 368.1856 [M + H]^+^, found: 368.1856.

**6m:***2-(4-(butylamino)-3,5-dimethylphenyl)-5,7-dimethoxy-4H-chromen-4-one*: Yellow powder; yield 70%; mp: 143.9–145.2 °C; Log *P*: 4.53; ^1^H-NMR (600 MHz, CDCl_3_) *δ*(ppm): 7.47 (s, 2H, Ar–H), 6.55–6.56 (m, 2H, Ar–H and C=CH), 6.34 (d, *J* = 2.4 Hz, Ar–H), 3.93 (s, 3H, OCH_3_), 3.90 (s, 3H, OCH_3_), 3.14 (t, *J* = 7.2 Hz, 2H, NHC*H*_2_), 2.32 (s, 6H, ArCH_3_), 1.53–1.58 (m, 2H, CH_2_), 1.37–1.43 (m, 2H, CH_2_), 0.94 (t, *J* = 7.8 Hz, CH_3_); ^13^C-NMR (151 MHz, CDCl_3_) *δ*(ppm): 177.8, 163.8, 161.3, 160.8, 159.9, 149.8, 127.5, 126.8, 122.9, 109.3, 107.1, 96.0, 92.8, 56.4, 55.7, 47.8, 33.4, 20.2, 19.3, 13.9; HRMS (ESI) calcd. C_23_H_28_O_4_N: *m*/*z* 382.2013 [M + H]^+^, found: 382.2012.

**6n:***2-(4-((2-fluorophenyl)amino)-3,5-dimethylphenyl)-5,7-dimethoxy-4H-chromen-4-one*: Yellow powder; yield 70%; mp: 125.9–130.1 °C; Log *P*: 5.42; ^1^H-NMR (600 MHz, CDCl_3_) *δ*(ppm): 7.61 (s, 2H, Ar–H), 7.05–7.08 (m, 1H, Ar–H), 6.87 (t, *J* = 4.2 Hz, 1H), 6.70–6.73 (m, 1H, Ar–H), 6.63 (s, 1H, C=CH), 6.57 (d, *J* = 2.4 Hz, Ar–H), 6.36 (d, *J* = 2.4 Hz, Ar–H), 6.23–6.26 (m, 1H, Ar–H), 5.45 (s, 1H, NH), 3.94 (s, 3H, OCH_3_), 3.91 (s, 3H, OCH_3_), 2.26 (s, 6H, ArCH_3_); ^13^C-NMR (151 MHz, CDCl_3_) *δ*(ppm): 177.7, 164.1, 160.9, 160.6, 159.9,151.8 (d, *J* = 293.64 Hz), 140.5, 136.0,133.5 (d, *J* = 11.02 Hz), 128.5,126.3, 124.4 (d, *J* = 3.47 Hz), 118.6 (d, *J* = 6.64 Hz), 115.0 (d, *J* = 18.42 Hz), 113.9 (d, *J* = 1.96 Hz), 109.2, 108.6, 96.2, 92.8, 56.4, 55.8, 18.6; HRMS (ESI) calcd. C_25_H_23_O_4_NF: *m*/*z* 420.1606 [M + H]^+^, found: 420.1603.

### 3.3. Method of Bioactivity Study

#### 3.3.1. Cell Lines and Cell Culture

The human leukemic cell line (HEL) and the prostate cancer cell line (PC3) were obtained from Molecular and Cell Biology Research, Sunnybrook Health Sciences Centre, Toronto, Ontario, Canada. All cells were cultured in RPMI medium (high glucose) supplemented with 5% fetal bovine serum (FBS; HyClone, GE Healthcare, Australia) and maintained in a humidified incubator with 5% CO_2_ at 37 °C. When the growing cells reached approximately 60–80% confluence, they were treated with **6k**. DMSO was used as the control.

#### 3.3.2. Cell Viability Assay

The cytotoxicity of **6k** on HEL and PC3 was measured by the MTT method [19]. The cells were plated at a density of 1 × 10^4^/well in a 96-well plate and incubated at 37 °C for 24 h. The cells were then treated with the compounds of interest for 48 h. After 20 µL of MTT (3-(4,5-dimethylthiazol-2-yl)-2,5-diphenyl-2*H*-tetrazolium bromide) was added to each well, the cells were incubated for 4 h. The cells were then removed from the medium and DMSO (100 mL) was added. The cell viability was detected by measuring the absorbance at 490 nm on a plate reader (Bio Tek, Winooski, VT, USA). All experiments were repeated at least three times.

## 4. Conclusions

Seventeen new flavone derivatives were synthesized via *N*-amination in the 4′-position. The in vitro tumor growth inhibitory activities of all of the derivatives were assayed using the human cell lines HEL and PC3. In general, compounds **3**, **4**, **6f**, **6e**, **6b**, **6c** and **6k** demonstrated the most potent antiproliferative activities against the HEL and PC3 cell lines. Preliminary structure–activity relationship studies indicated that the flavone bind the methyl amino acid at the 4′-position, thereby increasing the anticancer activity. Our findings suggested that the configuration of the methyl amino acid had some effects on the biological activity of the derivatives. These results provide new insight into developing flavonoid-derived anticancer agents. The antibacterial test showed that all of the derivatives did not possess obvious antibacterial activities.

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
