# Peer review of "Synthesis of Flavone Derivatives via N-Amination and Evaluation of Their Anticancer Activities"

_molecules, 2019, doi:10.3390/molecules24152723_

Reviewer 1 Report

The manuscript molecules-561767 is devoted to the actual problem of organic synthesis and medicinal chemistry. The reviewed article is interesting and theme of the article meets the scope of the journal. Work is performed at sufficient scientific level and has good quality; the results of investigation are professionally interpreted. However, it needs minor revision before publication.

To improve the quality and perception of the manuscript I would suggest paying attention to following comments:

Figure 1 should be removed. This is a well-known      structure that is presented in the textbooks on organic chemistry.

Page      2, line 41. “Andriy et al” must be “Golub et al”. Andriy is first name,      Golub is last name.

English language and style minor spell check      required

After correction this manuscript can be accepted for publication. My decision is minor revision.

Author Response

1)Figure 1 should be removed. This is a well-known structure that is presented in the textbooks on organic chemistry.

Answer: Figure 1 were removed.

2)Page 2, line 41. “Andriy et al” must be “Golub et al”. Andriy is first name, Golub is last name.

Answer: Andriy et al was modified to Golub et al.

English language and style minor spell check required

Answer: We have checked it carefully.

Reviewer 2 Report

This manuscript describes synthesis of 17 new compounds belonging to flavone group. I think that this manuscript should be interesting for the readers of Molecules, but the authors have to improve it.
1. Where is the conclusions of this manuscript?
2. Lack in the experimental part the bands of Infra Red value!
3. In abstract the authors described that "The structure-activity relationships evaluation showed that the configuration of methyl amino acid might affect their biological activities." and I didn't see the answer in Discussion  and Results. Why?
4. What is with solubility of the new compounds?

Author Response

1). Where is the conclusions of this manuscript?

Answer: The conclusions of this manuscript was added in row 326.

2). Lack in the experimental part the bands of Infra Red value!

Answer: Because of the limitation about the content of samples, the measurement of Infra Red value hasn′t been done for studying of anti-cancer activity. In addition, the current structural data (1H NMR, 13C NMR and HRMS) have been able to confirm the structure of new compounds.

3). In abstract the authors described that "The structure-activity relationships evaluation showed that the configuration of methyl amino acid might affect their biological activities." and I didn't see the answer in Discussion and Results. Why?

Answer: The discussion in this article is in row 303-307.

4). What is with solubility of the new compounds?

Answer: According to your suggestion, LogP were simulated by ChemDraw. At the same time, the value of LogP were provided before the NMR data for new compounds (3, 4, 6, 6a-6n).

Reviewer 3 Report

The manuscript entitled ‘Synthesis of flavone derivatives via N-amination and evaluation of their anticancer activities’ by

Ni Zhang, Jin Yang, Ke Li, Jun Luo, Su Yang, Jun-Rong Song, Chao Chen, and Wei-Dong Pan

reports the synthesis of a series of flavone derivatives and the evaluation of their antiproliferative activities against HEL and PC3 cell lines. For the most active compound 6k growth curves have also been reported. The topic is interesting, and this contribute adds further information about the anticancer activity of flavone derivatives.

The manuscript deserves some improvements/corrections, for example:

row 37; candidates, rows 49-52 need check, row 62 (perhaps the authors whould write Bruker?), row 68-69 and 74-75 a verb is missed.

13C NMR data of compound 6; authors wrote: 118.6 (dd, J = 320.12, 640.24, CF3): the multiplicity for CF3 is quartet, authors must correct (middle of the quartet for chemical shift and only one value for coupling constant).

Row 296: it is not clear why the number of active compounds are not sequential, as usually is reported. I suggest to ordinate them.

After the above indicated minor modifications, the manuscript is suitable for publication on Molecules.

Author Response

1) row 37; candidates, rows 49-52 need check, row 62 (perhaps the authors whould write Bruker?), row 68-69 and 74-75 a verb is missed.

Answer: Thank you very much. Candidate has been amended to candidates in row 37. In rows 49-52 there is a wrong sentence (“a series of was designed and synthesized a clase of by replacing of the 4′-OH with N-amino substituents” which have been changed to “a series of flavone dcrivatives were designed and synthesized by replacing of the 4′-OH with N-amino substituents.” In rows 68-69 and 74-75 a verb are added.

2) 13C NMR data of compound 6; authors wrote: 118.6 (dd,J= 320.12, 640.24, CF3): the multiplicity for CF3 is quartet, authors must correct (middle of the quartet for chemical shift and only one value for coupling constant).

Answer: The multiplicity for CF3 is quartet. We have made corrections as 118.6(q, J = 320.1, CF3)

3) Row 296: it is not clear why the number of active compounds are not sequential, as usually is reported. I suggest to ordinate them.

Answer: Thank you very much. According to opinion, active compounds are sequential in the table 1.